# PTHrP Regulates Fatty Acid Metabolism via Novel lncRNA in Breast Cancer Initiation and Progression Models

**DOI:** 10.3390/cancers15153763

**Published:** 2023-07-25

**Authors:** Rui Zhang, Jiarong Li, Dunarel Badescu, Andrew C. Karaplis, Jiannis Ragoussis, Richard Kremer

**Affiliations:** 1Research Institute, McGill University Health Center, Montreal, QC H4A 3J1, Canada; 2Department of Human Genetics, McGill University Genome Centre, McGill University, Montreal, QC H3A 0G1, Canadaioannis.ragoussis@mcgill.ca (J.R.); 3Lady Davis Institute for Medical Research, Montreal, QC H3T 1E2, Canada; andrew.karaplis@mcgill.ca

**Keywords:** PTHrP, RNA-sequencing, PyMT mouse model, breast cancer, LncRNAs, Scd1

## Abstract

**Simple Summary:**

The 5-year survival rate for women with metastatic breast cancer is 29%. Potential biomarker identification is important for new treatment modalities in affected patients. Previous studies have shown that Parathyroid hormone-related peptide (PTHrP) plays a critical role in breast cancer growth and metastasis. Our study aimed to use a genetically modified breast cancer mouse model to precisely examine the role of PTHrP from early primary breast cancer initiation to late progression. We identified a novel long non-coding RNA (lncRNA), a new target for fatty acid metabolism that can be regulated via PTHrP in our unique mouse breast cancer model. We confirmed that a potential human lncRNA, OLMALINC, plays a similar role in fatty acid metabolism that can be regulated via PTHrP and validated our mouse findings in human breast cancer cell lines. Genetically engineered mouse models provide valuable tools to study the molecular metabolism for breast cancer progression.

**Abstract:**

Parathyroid hormone-related peptide (PTHrP) is the primary cause of malignancy-associated hypercalcemia (MAH). We previously showed that PTHrP ablation, in the MMTV-PyMT murine model of breast cancer (BC) progression, can dramatically prolong tumor latency, slow tumor growth, and prevent metastatic spread. However, the signaling mechanisms using lineage tracing have not yet been carefully analyzed. Here, we generated Pthrpflox/flox; Cre+ mT/mG mice (KO) and Pthrpwt/wt; Cre+ mT/mG tumor mice (WT) to examine the signaling pathways under the control of PTHrP from the early to late stages of tumorigenesis. GFP+ mammary epithelial cells were further enriched for subsequent RNA sequencing (RNAseq) analyses. We observed significant upregulation of cell cycle signaling and fatty acid metabolism in PTHrP WT tumors, which are linked to tumor initiation and progression. Next, we observed that the expression levels of a novel lncRNA, GM50337, along with stearoyl-Coenzyme A desaturase 1 (Scd1) are significantly upregulated in PTHrP WT but not in KO tumors. We further validated a potential human orthologue lncRNA, OLMALINC, together with SCD1 that can be regulated via PTHrP in human BC cell lines. In conclusion, these novel findings could be used to develop targeted strategies for the treatment of BC and its metastatic complications.

## 1. Introduction

Nearly 30% of women initially diagnosed with early-stage breast cancer (BC) will present with metastases to distant organs, such as bones, lungs, liver, and brain, in addition to lymph nodes [1,2]. The 5-year survival rate for patients with localized BC is 90%; however, this survival rate drops to 28.1% in patients diagnosed with metastatic BC [2]. The biological processes underlying metastasis and therapeutic resistance are not yet fully understood, which hampers the development of personalized treatment interventions. In recent years, genome-wide analyses of BC have revealed that this is a heterogeneous disease comprising many subtypes with different treatment responses and clinical outcomes. Based on the presence or absence of the estrogen-receptor (ER), progesterone-receptor (PR), and human epidermal growth factor receptor 2 (HER2) status, BC can be classified into three clinical subtypes: hormone-receptor (HR) positive (HR+; ER+, PR+/−, and HER2−), HER2 positive (HER2+), and triple negative (TN; ER−, PR−, and HER2−) [3]. Five distinct molecular subtypes have been further defined based on gene expression profiling: luminal A, luminal B, HER2-enriched, basal-like, and normal-like. The luminal A subtype is a low-grade, slow-proliferating cancer, whereas the luminal B subtype is a more aggressive cancer with an accelerated proliferating rate [4]. Spontaneous BC mouse models are the most widely used because they can fully recapitulate the tumorigenesis from early- to late-stage metastasis. The most common genetic manipulations in BC models include the overexpression of several mammary oncogenes (e.g., PyMT, neu/ErbB2/HER2, Myc, Ras, SV40 Tag, cyclin D1, and Wnt1) or the specific deletion of tumor suppressor genes such as p53 [5].

PTHrP was initially identified as a humoral factor that underlies the development of hypercalcemia in malignancy, a severe complication in patients with advanced-stage cancers [6]. It is expressed in almost all normal tissues and involved in a wide range of normal developmental and physiological processes, including the regulation of proliferation and differentiation [7]. PTHrP is frequently overexpressed in a wide range of cancers, including breast, prostate, lung, colon, and pancreatic cancer [8,9]. Its expression has been shown to be under the control of numerous growth and angiogenic factors such as transforming growth factor beta (TGF-β), epidermal growth factor (EGF), platelet-derived growth factor (PDGF), and vascular endothelial growth factor (VEGF) [8]. Upon binding to the parathyroid hormone receptor 1 (PTHR1), PTHrP can activate intracellular cyclic 3′, 5′-adenosine monophosphate (cAMP), which further activates both the adenylyl cyclase/protein kinase A (PKA) pathway as well as the calcium/inositol phosphate/protein kinase C, AKT, Cyclin D1 as well as CREB and RUNX transcription factors [10,11,12,13,14]. Most human and animal studies have reported PTHrP as a pro-tumorigenic factor associated with a higher risk of metastasis in breast, lung, head and neck, lymphoma, prostate, and colon cancers [10,14,15,16,17,18,19]. However, some studies have linked PTHrP expression in BC patients to improved outcomes, and the ablation of PTHrP in a late-onset mammary cancer model has been shown to increase tumor progression and impair animal survival [20,21]. Current paradoxical results suggest an incomplete understanding of the role of PTHrP during tumor initiation and progression. How PTHrP fundamentally is involved in regulating these critical signaling pathways of BC initiation and progression remains elusive.

Next-generation sequencing (NGS) technology has revolutionized the field of cancer research over the last decade and a half. It unravels the fact that only about 2% of the human genome is annotated as protein-coding genes, whereas up to 98% of the transcriptional output is made of non-coding RNAs (ncRNAs) [22]. According to their length, ncRNAs can be further categorized as short ncRNAs (sncRNAs) of 18–35 nt in length, such as microRNAs (miRNAs), piwi-interacting RNAs (piRNAs), small interfering RNAs (siRNAs), and long non-coding RNAs (lncRNAs) ranging from 200 nt to 100 kilobases. Among them, lncRNAs are differentially expressed in tumors, and they are directly linked to the transformation of healthy cells into tumor cells [23]. Recent data show that lncRNAs can regulate several critical enzymes involved in lipid metabolism in which dysfunctions have been linked to oncogenic activation and the maintenance of cancer cell survival and metastatic potential [24]. For example, the Highly Up-Regulated In Liver Cancer long non-coding RNA (lncRNA-HULC) can activate the acyl-CoA synthetase subunit ACSL1, contributing to the malignant development of hepatocellular carcinoma [25]. In colon tumor cells, the lncRNA, UHRF1 protein-associated transcript (UPAT), together with the protein-associated transcript, Ubiquitin Like With PHD And Ring Finger Domains 1 (UHRF1), can up-regulate stearoyl-CoA 9-desaturase-1 (SCD1), which is a key enzyme for the control of lipid synthesis in cancer cells [26].

Our own work, using a mouse mammary tumor virus-polyoma middle tumor-antigen (MMTV-PyMT) oncoprotein mouse model of BC, has established that PTHrP is a key regulator of tumor initiation and progression [14]. We also demonstrated that PTHrP can regulate epithelial-to-mesenchymal transition (EMT) and cancer cell stemness during BC bone metastasis [27]. In this current study, we generated Pthrp*^flox/flox^*; Cre^+^ and Pthrp*^wt/wt^*; Cre^+^ mT/mG, PyMT BC mouse model, which allowed us to specifically trace and enrich the green fluorescent protein (GFP) mammary epithelial cells from early- to late-stage carcinoma. We applied comprehensive and unbiased transcriptome analyses to compare the gene expression profile between Pthrp*^wt/wt^*; Cre^+^ and Pthrp*^flox/flox^*; Cre^+^ tumors from three distinct stages of BC progression (hyperplasia, adenoma, and carcinoma). We demonstrate that the disruption of PTHrP expression has a significant impact on tumor growth mainly via its regulation of gene expressions involved in several cancer hallmark signaling pathways, including cell cycle and metabolic adaptation. Moreover, we provide novel insights into the molecular mechanism of PTHrP as an oncogene that can activate a key enzyme, SCD1, for fatty acid metabolism via a cis-acting mechanism of a nearby lncRNA. Therefore, we propose that targeting these novel pathways driven by PTHrP can be beneficial in the treatment of BC.

## 2. Materials and Methods

### 2.1. Mouse Lines and Breeding

All animals were housed in the Research Institute of the MUHC Animal Resources Division, and all animal protocols were reviewed and approved by the Glen Facility Animal Care Committee (FACC). MMTV-PyMT [28] and MMTV-Cre mice [29] on a pure FVB/NJ background were kindly supplied by Dr. William Muller (McGill University Cancer Center). C57BL/6 *Pthrp* floxed mice (*Pthrp^flox/flox^*) [30] were backcrossed 9 generations to FVB animals. Crossing *Pthrp* floxed or *Pthrp^wt/wt^* mice with PyMT-MMTV strain [28] or MMTV-Cre [29] mice (FVB background) produced PyMT-MMTV; *Pthrp^flox/flox^*; *Cre^+^* (knockout) and PyMT-MMTV; *Pthrp^wt/wt^*; *Cre^+^* (wild-type) tumor mice. ROSA mTmG reporter mouse line (mTmG) was obtained from the Jackson Laboratory (Bar Harbor, ME, USA). For lineage-tracing and RNAseq experiments, PyMT-MMTV; *Pthrp^flox/flox^*; *Cre^+^* and PyMT-MMTV; Pthrp *^wt/wt^*; *Cre^+^* mice were crossed with ROSA mT/mG mice.

### 2.2. Mice Genotyping

The genotyping of the PyMT, *Pthrp*, and *Cre* transgenes was performed via standard PCR using primer sets as described previously [14]. Genotyping was performed on DNA isolated using standard protocols from tail snips obtained at or just before weaning of litter. In brief, PCR reaction was carried out as follows: 94 °C for 30 s, 64 °C for 1 min, 72 °C for 1 min, and 35 cycles. The IVIS^®^ Spectrum in vivo imaging system was applied to measure the tail epifluorescence, which is sufficient to identify ROSA mTmG mice from wildtype mice.

### 2.3. HE Staining

Frozen tissues kept at −80 °C were fixed in 70% ethanol. Fixed tissues were stained with hematoxylin and eosin (H&E) by Leica Autostainer (Histopathology Platform in Research Institute of MUHC). Hematoxylin and eosin-stained permanent sections were examined using an Olympus IX51 microscope (Olympus, Tokyo, Japan) equipped with an Olympus DP71 camera (Olympus).

### 2.4. Confocal Microscopy

Fresh tissues were frozen at −80 °C and sectioned at 10 µm using a cryostat (Histopathology Platform in Research Institute of MUHC). Confocal images were acquired using a Zeiss LSM780-NLO Laser Scanning Confocal with IR-OPO lasers microscope, a 10×/0.30 WD = 5.2 objective, and Zen2012 image acquisition software (Molecular Imaging Platform in Research Institute of MUHC).

### 2.5. Isolation of Primary Mammary Gland and Breast Tumor

Primary tissues (tumor or gland) were isolated via stepwise mechanical disruption and enzymatic digestion according to our published protocols [14]. Tissues were harvested from 6- to 10-week-old mice, minced with a scalpel, and incubated for digestion for 2 h at 37 °C with gentle rocking in FBS-free DMEM (Multicell, Wisent Inc., St. Bruno, QC, Canada) supplemented with 2.4 mg/mL collagenase B and 5 U/mL dispase II (4942078001, Roche, Switzerland). Tissue fragments were washed with PBS, centrifuged, and resuspended in DMEM with 10% Fetal Bovine Serum (Gibco, US Origin, Waltham, MA, USA), 100 IU/mL penicillin, and 100 ug/mL streptomycin (Multicell, Wisent Inc., St. Bruno, QC, Canada).

### 2.6. FACS of Single GFP^+^ Cells

Breast tumor cells (from PyMT-MMTV; *Pthrp^flox/flox^*; *Cre^+^*; mTmG and PyMT-MMTV; Pthrp*^wt/wt^*; *Cre^+^*; mTmG mouse) or mammary epithelial cells (from *Pthrp^flox/flox^*; *Cre^+^*; mTmG and Pthrp*^wt/wt^*; *Cre^+^*; mTmG mouse) were harvested after 48 h incubation in DMEM with 10% FBS. The cells were washed in PBS, filtered through 40 μm cell strainers, and resuspended in complete DMEM before sorting. Following doublet exclusion using FSC-A and FSC-H scatter plot, GFP and/or tdTomato (mTd) expressing cells were sorted on a BD FACS Aria Fusion (BD Bioscience, Franklin Lakes, NJ, USA) at Immunophenotyping Platform in the Research Institute of MUHC. For the RNAseq experiments, a minimum of 20,000 single cells were sorted.

### 2.7. RNA Isolation and Sequencing

Total RNA was isolated either directly from the sorted GFP^+^ tumor cells or mammary epithelial cells after FACS using the miRNeasy mini kit (Qiagen, Venlo, The Netherlands) according to the manufacturer’s protocol. RNA quantity was assessed using Qubit 4 Fluorometer (Invitrogen, Montréal, QC, Canada), and RNA quality was evaluated using Agilent TapeStation (Agilent Technologies, Waldbronn, Germany).

For high-throughput sequencing, RNA samples were required to have an RNA integrity number (RIN) ≥ 7. The RNAseq transcriptome strand library was prepared by following the TruSeqTM Stranded Total RNA kit from Illumina (San Diego, CA, USA) using 100 ng of total RNA for each sample. Briefly, ribosomal RNA (rRNA) depletion was achieved using an Illumina Ribo-Zero Magnetic kit. First-strand cDNA synthesis used Super strand synthesis Act D kit containing random primers and Super Script II reverse transcriptase to convert the original mRNA to complementary cDNA. Second-strand cDNA synthesis involves the removal of the original RNA template and synthesizes a replacement strand, incorporating dUTP in place of dTTP to generate ds cDNA. After cDNA synthesis, a single “A” nucleotide is added to the 3′ ends of the blunt fragments to prevent them from ligating to one another during the adapter ligation reaction. During the adapter ligation process, each sample has a different RNA adapter index for identification in subsequent data analyses (Illumina^®^ Nextera™ DNA Unique Dual Indexes, San Diego, CA, USA). AMPure XP beads were used to purify the ds cDNA from the second strand reaction mix and leaving exclusively the blunt-ended cDNA for the subsequent DNA enrichment. The PCR was performed with a PCR primer cocktail that anneals to the ends of the adapters to exclusively enrich and amplify those DNA fragments that have adapter molecules on both ends. Only the first strand cDNAs were amplified in this step. After quantification using LightCycler^®^ 96 Instrument with KAPA library quantification kit, the paired-end RNA-seq library was sequenced with Illumina NovaSeq6000 at McGill Genome Center according to the standard protocol.

### 2.8. RNA-Seq Analyses

Sequences were aligned using Hisat first against the mouse transcriptome as defined by Gencode gene models M16 [31], with default parameters and the remaining unmapped genes to the Ensembl GRCm38 reference mouse genome [32]. Aligned reads from multiple read groups belonging to the same sample were indexed, sorted, and merged using sambamba v0.5.4 [33], a faster implementation of the Samtools algorithms [34]. Amplification duplicates were removed using Picard tools v1.128. Various quality controls from the RNASEQC package were used, including the genes detected, mapping rates, duplication rates, and intronic rates, based on metrics collected for each sample used [35]. HTSeq Count [36] was applied to enumerate the counts for each gene using the Gencode M16 GTF. All statistical analyses were carried out with R v3.6.2 [37]. We performed minimal pre-filtering, keeping only genes having at least 10 reads. Differential expression and principal component analysis (PCA) were performed using Deseq2 and R standard packages [38]. 

### 2.9. Time-Course Data Analyses

To find gene expression trends along tumor initiation and progression and detect the fundamental differences between PTHrP WT and KO tumors, the means of normalized expression reads of 1193 DEGs from three PTHrP WT 6-week tumor samples (compared to two 6-week WT normal gland samples) and 572 DEGs from three PTHrP KO 8-week tumor samples (compared to two 8-week KO normal gland samples) were used as an input to analyze the temporal expression patterns during BC progression including hyperplasia, adenoma and carcinoma stages. We then used k-means clustering to group these DEGs based on their normalized mean expression. We plotted the number of clusters against the total within-cluster SS (sums of squares), and K means = 3 was selected as the optimal one. GSEA (Gene Set Enrichment Analyses) was used to investigate hallmark gene sets (“Hallmark gene sets summarize and represent specific well-defined biological states or processes and display coherent expression”, as defined by GSEA) [39].

### 2.10. Correlation Analyses between lncRNA and Protein Coding Genes

The cis-regulatory target protein-coding genes by lncRNAs were identified via the following procedures. For each lncRNA, we identified protein-coding genes as “cis-regulated protein-coding genes” when (1) the mRNAs locus is within 500 k windows up- and downstream of the given lncRNA, and (2) the Pearson correlation of lncRNA-protein coding genes expression is significant (*p*-value of correlation less than 0.01). For each pair, Pearson correlation was performed to assess the correlation. Pearson correlation coefficients were calculated using cor() function, and *p* values were obtained using corr.test() from library(‘psych’) in R.

### 2.11. Cell Culture

Murine 4T1 and E0771 BC cells provided by Dr. Shafaat Rabbani (Department of Medicine, McGill University, Montréal, QC, Canada); human MDA-MB-231 and BT549 BC cells obtained from ATCC (Manassas, VA, USA) were cultured in DMEM (Multicell, Wisent Inc., St. Bruno, QC, Canada) supplemented with 10% FBS (Gibco, US Origin), 100 IU/mL penicillin, and 100 ug/mL streptomycin (Multicell, Wisent Inc., St. Bruno, QC, Canada). All cell lines were cultured in a cell culture incubator at 5% CO_2_ at 37 °C. Mouse anti-PTHrP monoclonal antibodies PA158 used in this study is a PTHrP-specific mouse monoclonal antibody (mAb) (subclass IgG3) against hPTHrP1–33 peptide. This mAb is highly specific: no reaction with PTH and no cross-reactivity between antibodies and other fragments of PTHrP as described in US patent # US78977139B2. 

### 2.12. CRISPR/Cas9 Genetic Knockout

To generate a genetic knockout of human Pthlh, a U6G RNA-Cas9-2A-redfluorescence protein vector (Invitrogen, Montréal, QC, Canada) was constructed with a human Pthlh-specific guide RNA directed to exon 4 for knockout/KO) or an empty vector for control (Pthlh^WT^ cells used as controls were transfected with the empty vector). The single guide RNA (sgRNA) sequence CGTCGCCGTAAATCTTGGATGG was inserted into the vector pCMVcas-9-RFP. The plasmid sequence was confirmed by sequencing at Genome Québec, McGill University (Montréal, QC, Canada). MDA-MB-231 cells grown to 40% to 80% confluency in antibiotic-free DMEM medium with 10% FBS were transfected with 2–3 μg of DNA using the PLUS transfection reagent (Thermo Fisher, Waltham, MA, USA). Cells were grown for 24 h, trypsinized, and selected via fluorescence-activated cell sorting (FACS) (FACS-Aria II cell sorter: BDBiosciences, Mississauga, ON, Canada) for red fluorescent protein. Single clones were isolated using an autoMACS Pro Separator (Miltenyi Biotec, Somerville, MA, USA) and expanded in single wells (96-well plate). DNA sequencing was conducted on isolated clones to confirm knockout status. This protocol has been described in detail in our previously published paper [27]. 

### 2.13. Quantitative Real-Time PCR (qRT-PCR) Assays

Single-stranded cDNA was synthesized from 1 µg of total RNA using 5X All-In One RT MasterMix Kit (Abm) followed by qPCR with PowerUp SYBR Green Master Mix (Applied Biosystems, Foster City, CA, USA) on a ViiA 7 Real-Time PCR System (Applied Biosystems). A standard program for the qPCR thermocycler was as follows: 10 min at 95 °C for the initial activation step followed by 40 cycles of 95 °C for 15 s, 52 °C for 15 s for denaturation, and 72 °C for 1 min annealing/elongation, and with subsequent melting curve analyses. Mouse Gapdh and Hprt were used as endogenous controls to normalize each sample. The samples were run in triplicate. Primer sequences are provided in Appendix A.

### 2.14. Antisense Oligonucleotide (ASO)-Mediated Knockdown (KD)

Mouse primary tumor cells were seeded at a density of 500,000 cells/well (1000 mL per well) into 6-well plates. Transfection-free uptake of ASOs was accomplished by adding 5 µM of either a lncRNA-specific ASO or scrambled ASO (scASO) to the primary cell culture medium immediately after seeding the cells. The plates were incubated in a 5% CO_2_ incubator at 37 °C for indicated time points, and RNA was isolated using the RNeasy 96 kit (QIAGEN) according to the manufacturer’s instructions. qRT-PCR was performed to check the knockdown efficiency. ASO sequences are provided in Appendix A.

### 2.15. Cell Viability Assays

A day prior to treatment with ASOs, PyMT primary tumor cells were cultured for adherence in 96-well plates in DMEM media with 10% FBS at a density of 15,000 cells/well (100 μL per well) at 37 °C under 5% CO_2_. The growth media was aspirated and overlayed with new media containing 10% FBS and 5 μM of either a lncRNA-specific ASO or scASO. After gentle mixing, the cells were grown for 72 h at 37 °C. A 5% final concentration of PrestoBlue^®^ was added to each well and incubated for 2 h at 37 °C. Fluorescence intensity (top-read) was measured using a multi-well plate reader (Tecan Infinite 200 Pro, Tecan, Männedorf, Switzerland) with excitation at 560/10 nm and emission at 590/10 nm at a 50% fixed gain.

### 2.16. Cell Invasion Assay

Briefly, biocoat Matrigel invasion champers were from BD Bioscience (Cat354480), PyMT primary tumor cells were tested for Matrigel invasion for 36 h with GM50337-ASO 5 μM, scASO 5 μM or 5 μM Scd1 inhibitor and DMSO as described earlier [8]. Invading cells were counted after paraformaldehyde fixation and toluidine blue staining.

## 3. Results

### 3.1. Mouse Model for Fluorescence-Based Mapping of Mammary Epithelium

To generate mice in which the mammary epithelium-expressing membrane-targeted GFP is highlighted against the membrane-targeted red fluorescent backlight of stromal and nonepithelial-derived mammary gland tissues, we used our previously generated strains, including *Pthrp^wt/wt^*; *Cre^+^* tumor-free (PTHrP WT tumor-free), *Pthrp^wt/wt^*; *Cre^+^* tumor (PTHrP WT tumor), *Pthrp^flox/flox^*; *Cre^+^* tumor-free (PTHrP KO tumor-free), and *Pthrp^flox/flox^*; *Cre^+^* tumor (PTHrP KO tumor), that were crossed with mTmG mice. This process gave rise to four different genotypes of inbred mice (Figure 1A). 

In mammary epithelial cells where Cre recombinase is expressed (i.e., in the PTHrP WT and KO tumor-free), the mTd gene is excised, resulting in a loss of mTd fluorescence and a gain of mGFP expression. Accordingly, 3D and confocal fluorescent microscopy of normal mammary glands from tumor-free mice confirmed that all the ducts and alveoli of the mammary gland were GFP^+^, as expected (Figure 1B,C), whereas nonepithelial-derived mammary gland tissues such as adipose and connective-tissues (fibroblasts) were mTd^+^ (Figure 1B,C). Consistently, BC cells isolated from tumor mice were GFP^+^ from the early hyperplasia until the late carcinoma stage, indicating that BC cells are of epithelial origin (Figure 1C). To assess for unanticipated toxic effects of fluorescent protein (mTd or GFP) expression in the mammary gland, we performed histological staining of the mammary gland from tumor and tumor-free bearing animals. The histological appearance of the mammary gland was unremarkable using H&E-stained sections from these mice (Figure 1C). In line with our previous study, we observed no detectable differences in mammary gland development and lactation capacity following *Pthrp* ablation in female mice crossed with mTmG mice [14].

Furthermore, both mammary epithelial cells and BC cells could be identified as GFP positive via flow cytometry. Flow cytometry analyses of digested mammary tissues from tumor and tumor-free animals could clearly separate GFP+ cells from mTd+ cells (Appendix A). It should also be noted that a small number of cells were double positive (expressing both GFP and mTd). This may have been the result of cells in which the expressed mTd protein after Cre excision has not yet been completely removed or from recombination occurring in only one mTmG allele (but not both) in homozygous mice. Thus, the specific expression of GFP enabled us to purify mammary epithelial cells and breast tumor cells via fluorescence-activated cell sorting (FACS) for the subsequent in vitro RNAseq analyses.

### 3.2. Pthrp Ablation Modifies Fatty Acid Metabolism as well as Cell Cycle Events in BC Initiation and Progression 

To determine the transcriptional distinction between PTHrP WT and KO tumors from early hyperplasia to late carcinoma stages, we collected mammary tumors from three animals per group (PTHrP WT and KO tumor mice) and normal mammary glands from two animals per group (age-matched PTHrP WT and KO tumor-free) at 6-, 8-, 10-, and 12-week time points, respectively (Appendix A). We obtained an average of ~170 million total number of reads for each sample from our RNAseq data (Appendix A). An average of 24,792 genes were detected in each sample with an 86% mapping rate and 41% duplication rate (Appendix A). We validated the origin of the cells by assessing the presence of GFP (recombined) or mTd (non-recombined) transcripts (reads were aligned to the mouse genome merged with GFP and mTd spike-in sequences). We found that GFP is differentially expressed at higher levels compared to mTd in all our samples, indicating that we had exclusively sequenced the cells of epithelial origin (Appendix A). We also confirmed that none of our PTHrP KO samples had aligned reads at the fourth exon of the *Pthlh* gene (loxP sites), indicating that enriched GFP^+^ cells show a good concordance with Cre expression in mammary epithelium, leading to a precise excision of the loxP-flanked region in the *Pthlh* gene (Appendix A). Our RNAseq analysis pipeline is highlighted in the figure (Figure 2A).

First, we performed PCA analyses (Method). A total of 32 dots are visible in Figure 2B, corresponding to 32 samples. Based on these analyses, tumor samples were clearly clustered and separated from the tumor-free samples (PC1), indicating proper sample processing with high-quality sequencing results. Moreover, we found that the most striking differences observed between PTHrP WT and KO samples (the different plotting shapes) are considerable (PC2), though not stronger than the differences between tumors and normal glands (PC1). DEGs were examined at each time point by comparing three tumors and two age-matched tumor-free samples. The analyses identified 1741, 2313, and 1582 DEGs in the PTHrP WT tumors at weeks 6, 8, and 10, respectively, and the numbers of DEGs in the PTHrP KO tumors were 916, 1576, and 1478 at weeks 8, 10, and 12, respectively (fold change > 1.5, false discovery rate (FDR) < 0.1, Figure 2C). We also found that the number of down-regulated DEGs was higher than the number of up-regulated DEGs at each stage in WT and KO tumors (Figure 2C). Among the union of all DEGs (3271 for PTHrP WT and 2282 for PTHrP KO), a significant proportion of DEGs (24%, 778 for WT and 26%, 590 for KO) appeared at all three time points (Figure 2C).

To further identify the fundamental difference between PTHrP WT and KO tumors, GSEA was applied to investigate hallmark genes. Of the GSEA hallmarks, remarkable gene expression differences between PTHrP WT and KO during the three stages have been highlighted (Figure 3). Key pathways linked to tumor initiation and progression, including G2M checkpoint, DNA repair signaling, and fatty acid metabolism, were activated in PTHrP WT tumors as early as 6 weeks (Figure 3A). In contrast, the pathway regulating apoptosis, a key pathway associated with the inhibition of tumor progression, was activated in PTHrP KO tumors (Figure 3B).

Next, we focused on the PTHrP-regulated genes that are differentially expressed and might act as tumor drivers at the hyperplasia stage. We used the tumor-free mice as a reference and identified 1193 differentially expressed genes (week 6) in PTHrP WT tumors and 572 genes (week 8) in KO tumors (fold change > 2, false discovery rate (FDR) < 0.1). We used these DEGs from both WT and KO tumors as input data to analyze the temporal expression patterns during BC progression. Using k-means clustering, we then separated these DEGs into three clusters (Figure 4). In cluster 1, 160 genes from PTHrP WT and 50 genes from KO tumors rapidly increased in expression at the hyperplasia stage but gradually decreased as the tumor progressed, suggesting an important role in tumor initiation (Figure 4A,B and Appendix A). In cluster 2, 206 DEGs from the WT tumor and 197 DEGs from the KO tumor gradually increased in expression and maintained a high level throughout tumor progression, supporting their functional roles during tumor progression (Figure 4D,E and Appendix A). In cluster 3, 827 DEGs from the PTHrP WT tumor and 352 DEGs from the KO tumor had a dramatically reduced expression at hyperplasia and further maintained a low expression as the tumor progressed (Appendix A).

A subsequent comparison of the gene expression differences between the PTHrP WT and KO tumors revealed, in cluster 1, 157 DEGs were exclusively overexpressed in PTHrP WT, whereas 47 DEGs overexpressed solely in KO tumors (Figure 4A,B). Kyoto Encyclopedia of Genes and Genomes (KEGG) analysis was performed on these 157 DEGs from the PTHrP WT tumors. Among these, Ccnb1, Ccnb2, Cdk1, and Ccna2, mainly function in the cell cycle (KEGG: 04110) and maintain their highest level of expression in early hyperplasia. GSEA was also performed on all genes because of the inherent limitations of DEG analysis. Consistently, we observed remarkable differences in cell cycle signaling pathways by comparing PTHrP WT and KO tumors in the early stage. G2M checkpoint and E2F targets were upregulated in PTHrP WT tumors as early as 6 weeks but not upregulated in the early stage of KO tumors (8 weeks) (Figure 4C). Our transcriptomic results identified these specific DEGs from cluster 1 and signaling pathways from GSEA for BC initiation, which are driven by PTHrP. Our results further support our own and other previously published studies in which PTHrP was shown to regulate the cell cycle [7,14].

Interestingly, 118 DEGs are exclusively overexpressed in WT tumors driven by PTHrP in cluster 2 (Figure 4D,E and Appendix A). Among these, Scd1, which is a key enzyme in the de novo synthesis of fatty acids (FAs), was steadily upregulated in PTHrP WT tumors. Of the GSEA hallmarks, we uncovered the remarkable activation of fatty acid metabolism signaling pathways in PTHrP WT tumors as early as 6 weeks, but no changes were observed in the early stage of KO tumors (Figure 4F). Hereby, we demonstrated for the first time that metabolic changes can be associated with PTHrP ablation using RNA sequencing and observed, in PTHrP KO tumors, a significant delay of such lipidomic remodeling, widely accepted as a metabolic hallmark of BC progression. We also observed that 247 DEGs overlapped between PTHrP WT and KO tumors in cluster 3 mainly associated with varieties of cellular functions (Appendix A). These genes are specifically implicated in the cGMP-PKG signaling pathway (KEGG: has04022) and include Trpc6, Myl9, Prkg1, and Kcnmb1 (Appendix A). We also noticed decreased gene expression in other important signaling pathways, including Wnt (KEGG: hsa04310) and Hippo signaling pathway (KEGG: hsa04390).

### 3.3. Pthrp Ablation Deregulated lncRNA Expression in BC Initiation and Progression 

Remarkably, we observed ablation of PTHrP not only interrupted the protein-coding genes (PCGs) expression but also lncRNAs expression. As early as the hyperplasia stage, there are 110 PCGs (70%) and 44 lncRNA genes (28%) from cluster 1, 79 PCGs (67%) and 38 lncRNA genes (32%) from cluster 2, and 479 PCGs (83%) and 97 lncRNA genes (17%) from cluster 3 that are exclusively differentially expressed in PTHrP WT tumor compared to tumor-free controls (Appendix A). Therefore, we identified a total of 179 lncRNAs that were exclusively differentially expressed in PTHrP WT tumors in hyperplasia.

LncRNAs often regulate the expression of cis- (neighboring) or trans- (distal) PCGs. Therefore, the potential interactions between the lncRNAs and nearby PCGs are useful to examine the biological functions of lncRNAs. To explore the potential cis-regulatory functions of these lncRNAs, we performed Pearson analyses to evaluate the correlation of 179 lncRNAs and 668 PCGs that are differentially expressed. We further mapped these lncRNAs and PCGs to their genomic loci with a Pearson correlation coefficient of no less than 0.80 and a *p* value less than 0.01. LncRNAs located within 500 K ranges upstream or downstream of the PCGs were further screened. A total of six pairs of upregulated lncRNA-PCG were obtained from the lncRNA-PCG co-expression analyses (Table 1). The correlation analyses indicate that lncRNAs, like PCGs, may play significant roles in oncogenic pathways. We are mostly interested in lncRNA targets because PTHrP mediates cancer cell metabolism and regulates tumor initiation and progression via lncRNA can be a novel mechanism. Out of the six pairs, we next focused on the *GM50337-Scd1* pair from cluster 2, which supports the role of lncRNAs involving cellular metabolism via the regulation of their nearby PCG in a cis-regulatory mode.

### 3.4. Interruption of GM50337-Scd1 Decreases Cell Viability

In order to validate the relevant role of the *GM50337*-*Scd1* regulated by PTHrP in our mouse RNAseq analyses, we next performed in vitro functional studies by knocking down this novel lncRNA in primary mouse BC cells. *GM50337* gene is on mouse chromosome 19 (strand+) and encodes a single transcript with 1023 nucleotides containing four exons, and it is located adjacent to *Scd1* (Figure 5A). qPCR confirmed that *GM50337* is overexpressed in mammary tumor tissues compared to normal mammary glands (Appendix A). Based on the computational coding potential prediction programs, *GM50337* RNA transcripts have very low protein-coding potential and are evidenced as noncoding RNA (Appendix A) [40,41]. Our correlation analyses suggested that *GM50337* can potentially *cis*-regulate nearby *Scd1* and might contribute to BC progression. We tested this possibility by performing knockdown experiments of *GM50337* using antisense oligonucleotides (ASOs) in PyMT primary tumor cells. These oligonucleotides are short, single-stranded DNA molecules containing phosphorothioate-modified nucleotides. Importantly, we found that the uptake of ASOs in primary mammary tumor cells is efficient without the use of transfection agents, as reported previously [42]. We tested three different specific ASOs listed in the methods and achieved knockdown efficiencies at more than 50% after 24 h incubation using 5 µM of the most potent ASO (Figure 5B). Concomitantly, the knocking down of *GM50337* resulted in a 40% decrease in *Scd1* expression (Figure 5B), indicating that *GM50337* can *cis*-regulate *Scd1* expression.

To further investigate the functional impact of *GM50337* downregulation on tumor cell viability, we examined the ASO treatment on cell viability using PrestoBlue assay. We observed a 50% decrease in cell viability after 72 h of *GM50337*-specific ASO treatment but no effect with a scASO control (Figure 5C). A similar effect on cell viability was observed in 4T1 and E0771 mouse BC cells (Appendix A). To exclude the possibility of an off-target effect of ASO due to complementary binding between the ASO and unintended RNA with a sequence similar to *GM50337*, we evaluated cell viability by treating the cells with an Scd1 inhibitor (CAY10566, Cayman Chemicals, Ann Arbor, MI, USA). We observed a significant decrease in cell viability after 72 h treatment of Scd1 inhibitor, similar to the effect of ASO treatment (Figure 5D). Reductions in cell viability were also observed in 4T1 and E0771 mouse BC cells treated with the Scd1 inhibitor (Appendix A). In an invasion assay using a Matrigel matrix-coated chamber, the number of *GM50337*-specific ASO-treated PyMT primary tumor cells penetrating through the matrix was significantly decreased than that of control cells (Figure 5E,F). We also observed a significant decrease in cells penetrating through the Matrigel after 36 h with Scd1 inhibitor, similar to the effect of ASO treatment (Figure 5E,F). Our in vitro results validated the relevance of PTHrP-linked *GM50337* overexpression to tumor cell growth.

### 3.5. PTHrP Regulates BC initiation via Fatty Acid Metabolism

To confirm the relevance of the mouse lncRNA in human BC, we compared mouse and human lncRNA transcripts on the level of sequence conservation or location conservation as a potential regulatory of SCD. Many lncRNAs are conserved between different species on the level of genomic location rather than based on sequence, which could imply functional conservation [43]. Our primary analyses showed no sequence conservation of *GM50337* in the human genome. We, therefore, extended our search for the orthologs by analyzing the neighboring lncRNA of the same PCG, SCD, which we found is highly conserved among human and mouse genomes. Here, we focused on a human lncRNA, OLMALINC, which resides immediately downstream from the SCD gene on human Chromosome 10 with 988 nucleotides containing 3 exons (Figure 6A). A previous study has shown that OLMALINC is an enhancer of SCD transcription by forming a DNA–DNA looping interaction in human hepatocytes [44]. Two active transcription start sites (TSSs) have previously been confirmed in the enhancer and promoter of OLMALINC in HepG2 cells by GRO-seq data [44]. In line with these findings, we found a high correlation between OLMALINC and SCD in BC with an R-value of 0.45 using the Gepia2 tool from the TCGA database (http://gepia.cancer-pku.cn/ (accessed on 1 January 2022)) (Appendix A) [45]. Taken together, these findings suggest that OLMALINC could be the human orthologue of mouse *GM50337* and can regulate its adjacent regional SCD. 

To assess the effects of the PTHrP on cellular lipid metabolism via OLMALINC gene expression, we treated two different human BC cell lines with a PTHrP monoclonal antibody (mAb) which was generated in our laboratory. In line with our mouse RNAseq results, a 70% to 80% decrease in OLMALINC expression following PTHrP mAb treatment for 72 h resulted in an approximately 70% decrease in SCD expression in both MDA-MB-231 and BT-549, respectively (Figure 6B). As a validation of the PTHrP mAb treatment effect on human BC cell lines, we proceeded to generate a knockout clone of PTHrP in MDA-MB-231 BCs using CRISPR/Cas9 system (Appendix A). Loss of PTHrP significantly reduced both OLMALINC and SCD expression, recapitulating the PTHrP mAb treatment (Appendix A). Our results indicate that overexpression of GM50337-Scd1 identified from our PyMT mouse model unravels a specific human lncRNA target, OLMALINC, which has the potential to impact human BC initiation and progression via PTHrP regulation.

Moreover, OLMALINC has been identified to contain sterol regulatory element binding protein 1/2 (SREBP1/2) binding sites at its TSSs using ENCODE project data [44]. SREBP1/2 is a transcriptional factor, which can activate a series of genes for fatty acid synthesis, including ATP citrate lyase (ACLY), acetyl-CoA carboxylase (ACC), fatty acid synthase (FASN), and SCD [46]. Several critical signaling pathways, such as phosphatidylinositol 3-kinase (PI3K)/protein kinase B (PKB, Akt)/mammalian target of rapamycin (mTOR), epidermal growth factor receptor (EGFR), and Ras, can regulate SREBP1/2 activation to mediate tumor growth and progression [46]. Our previous paper highlighted that PTHrP can activate the PI3K/AKT pathways, which drive cell growth and survival [8]. Therefore, we hypothesized that OLMALINC and SCD expression can be regulated via PTHrP through SREBP1/2 activation. First, we confirmed a high correlation between SREBP1 and OLMALINC in BC with an R-value of 0.53 using the Gepia2 tool from the TCGA database (http://gepia.cancer-pku.cn/ (accessed on 1 January 2022)) (Appendix A) [45]. Using RT-qPCR, we showed that PTHrP mAb treatment for 72 h results in a significant decrease in SREBP1 expression for both MDA-MB-231 and BT-549, respectively (Figure 6C). We also showed that FADS2, a rate-limiting enzyme for an alternative fatty acid desaturation pathway, has a significant decrease in expression after 72 h treatment using PTHrP mAb (Figure 6C). These data suggest that OLMALINC-SCD expression is responsive to PTHrP mAb treatment, which can be a major underlying mechanism for regulating fatty acid metabolism in BC.

## 4. Discussion

We generated a mammary epithelial-specific Pthrp-knockout in PyMT mice in which mammary epithelial cells are specifically traced by membrane-targeted GFP. In line with our previous findings using the same BC mouse model, we observed genes that regulate cell cycle signaling were significantly upregulated by PTHrP as early as hyperplasia. We also demonstrated for the first time that PTHrP upregulates Scd1 encoding for the key fatty acid metabolism enzyme via the activation of a novel oncogenic lncRNA-GM50337. We further validated our findings in human BC cell lines demonstrating that PTHrP can regulate the transcription of human SCD via a candidate human lncRNA. Overall, we used an unbiased RNAseq approach to unravel all the transcriptomes altered via Pthrp ablation, including critical PCGs and novel lncRNAs regulating fatty acid metabolism. 

To investigate the involvement of PTHrP in primary BC initiation and progression, we generated a mammary epithelial-specific Pthrp-knockout (Pthrpflox/flox; Cre+) in a transgenic BC mouse model (PyMT) which Cre specifically expressed in mammary epithelium is monitored by membrane-targeted GFP (mT/mG reporter). During BC initiation, pre-neoplastic transformed epithelial cells only represent a small proportion of the total mammary epithelial population among all types of cells in the mammary gland. This advanced mouse model allowed us to specifically enrich total mammary epithelial populations, including both normal and pre-neoplastic cells for subsequent RNAseq analyses as early as hyperplasia. Therefore, we were able to specifically examine the transcriptomes of mammary epithelial lineage by comparing the tumor with tumor-free mice in order to identify the DEGs underlying epithelial phenotypes during tumor initiation using such a novel approach. Secondly, our previous studies have shown that lung metastasis from PTHrP KO mice was mainly derived from PTHrP-expressing tumor cells that escaped gene ablation [14]. This is a known limitation of the universal Cre/loxP model in which recombination (gene-deletion) efficiency is inconsistent [5]. This novel generated mouse model allowed us to isolate the complete PTHrP deletion (expressing GFP) from non-recombined (escapees) cells (expressing mTd) by FACS in mammary epithelial cells or BC cells. Our mouse model ensured that any transcriptomic or signaling pathway changes identified from RNAseq can be ascribed to the loss of the PTHrP gene in mammary epithelial cells from tumor and tumor-free animals. 

Even though this mouse model is an invaluable tool for our study, they have some limitations. Specifically, hierarchical clustering analyses indicated that the MMTV-PyMT mouse model is initially associated with the mature luminal signature characterized by the expression of luminal keratin 8/18, ER+, and PR+ at an early stage of tumor progression but progresses to luminal progenitor features characterized by basal-like tumors with low levels of ER in later stages, which is associated with metastasis [47]. In contrast, breast tumors in women are characterized by significant heterogeneity of individual tumor tissues as well as different pathological and molecular subtypes that have different treatment responses and clinical outcomes. Therefore, our results here cannot be extended to all types of human breast tissues but are more representative of basal-like. This may explain the apparent discrepancy of our results compared to Fleming et al. [14,20] in which Pthrp was disrupted in a very different model, the MMTV-neu mouse model, which was previously classified as luminal B BC [20]. 

In our study, we reported for the first time the transcriptional changes between PTHrP WT and KO tumors associated with three distinct stages in BC progression in mammary epithelial cells. Pthrp ablation was accompanied by a significant delay in tumor initiation characterized by the disruption of several hallmarks of cancer, including proliferation signaling and cellular metabolism. We observed that the ablation of Pthrp significantly downregulated genes involved in the cell cycle, including G2M checkpoint and E2F targets, supporting our previous finding using the same BC mouse model [14]. E2F transcription factors play critical roles in the cell cycle, and specifically high E2F score increases the expression of cell proliferation-related genes and is significantly associated with BC aggressiveness [48]. DEGs regulating cell cycle signals from cluster 1 exclusively overexpressed in PTHrP WT tumors further confirmed the critical role of PTHrP in the cell proliferation process during tumor initiation. Cell cycle and proliferation genes are also strong predictors of metastasis [49]. Many studies have confirmed the role of PTHrP in BC bone metastasis in the late stage. Future studies should focus on examining whether PTHrP-regulated genes in the early stage have a fundamental role in BC metastasis and how many signaling pathways have been involved in this process. 

Importantly, we observed, for the first time, using an unbiased genome-wide gene expression approach, that PTHrP can regulate fatty acid metabolism in BC early stage. Alterations of lipid metabolism have received increasing attention as a hallmark of cancer, and the dysregulation of its related enzymes has been linked to oncogenic activation and the maintenance of cancer cell survival and metastatic potential [24]. SCD1 is a key enzyme in the de novo synthesis of fatty acids, catalyzing the conversion of saturated fatty acids (SFAs) into monounsaturated fatty acids (MUFAs). Accelerated cancer cell proliferation is characterized by a greater demand for MUFAs, which are utilized mainly for the synthesis of membranes and signaling molecules. The accumulation of MUFAs, together with the overexpression of SCD1 in various types of tumors, such as lung, breast, prostate, colon, kidney, and lymphoma, has been observed in previous studies [50]. BC patients with high levels of tumor-derived SCD1 have significantly shorter survival rates, suggesting that an increase in SCD1 activity may correlate with tumor aggressiveness and poor patient prognosis [51]. We observed that the knockout of Pthrp remarkably reduced the expression of Scd1, and the interruption of its expression impaired BC growth. We also know from our previous study that Pthrp ablation significantly improves survival [14]. These results are consistent with other studies showing that the inhibition of SCD1 has been found to effectively suppress tumor cell proliferation and promote apoptosis in various types of tumors. Several in vitro studies demonstrated that the stable knockdown of SCD1 in cancer cells led to a decrease in the rate of cell proliferation and induce apoptosis via the depletion of MUFAs [52,53]. Further studies showed that SCD1 inhibitors suppressed the proliferation and induced the apoptosis of cancer cells of different origins, such as colon, lung, kidneys, and bladder [54,55,56,57]. Similar effects were observed in BC using another inhibitor of SCD1, namely CAY10566 [58]. In vivo mouse xenograft models illustrated that tumor growth is impeded by Scd1 shRNA knockdown and Scd1 pharmacological inhibition in renal cell carcinoma, endometrial cancer, glioblastoma, and prostate cancer [56,59,60,61]. In our study, we used the Scd1 inhibitor, CAY10566, and observed a significant decrease in the proliferation rate of PyMT primary cells, further supporting the role of Scd1 in our mouse model. 

Our work also provides additional support for future applications of anti-PTHrP therapy in clinical trials in patients with BC. Human BC cells treated with anti-PTHrP mAb in vitro had a remarkable decrease in growth [27]. We also showed that the anti-PTHrP therapy effect on tumor cell growth in vitro is mechanistically related to tumor cell lipid remodeling. We further identified that anti-PTHrP therapy significantly reduced SREBP1 expression along with SCD1 and FADS2. SREBP1 is regarded as the master transcriptional regulator of fatty acid metabolism and regulates the main lipid metabolism enzymes [24]. Bao et al. showed that SREBP1 can promote BC cells’ migration and invasion, and its upregulation has been proposed as a poor prognostic marker in BC patients [62]. Recently, ncRNAs have emerged as a master regulator in reshaping cellular metabolism in cancer, and LncRNA-HR1 was shown to repress the SREBP1 promoter activity, decreasing lipid metabolism in the Huh7 hepatocarcinoma cell line [63]. In our in vitro study, human BC cells treated with anti-PTHrP mAb not only downregulated genes coding for key enzymes in fatty acid metabolism but also disrupted OLMALINC expression, a human ortholog of mouse GM50337 identified in our RNAseq analyses and functionally validated in vitro using the mouse primary BC cells. However, whether PTHrP has a direct or indirect effect in regulating fatty acid metabolism in BC needs further investigation. 

Moreover, our RNAseq data revealed that lncRNAs are abundantly differentially expressed in mammary tumors compared to normal mammary epithelial cells as early as the hyperplasia stage. Several lncRNAs identified in our study have been studied previously. One of these lncRNAs is Foxd2os, and the corresponding human counterpart is lncRNA FOXD2 adjacent to opposite strand RNA 1 (FOXD2-AS1). FOXD2-AS1 is aberrantly expressed in various human cancers, including BC, and is associated with cancer progression [64]. Moreover, C130071C03Rik and A730020E08Rik were identified in an RNAseq screen on tumor organoids derived from PyMT mice and shown to be significantly overexpressed in breast tumors compared to normal breast tissue [42]. The human counterpart of C130071C03Rik is LINC00461, which is also significantly elevated in BC [42,65]. Emerging evidence has shown that lncRNAs play a critical functional role in BC initiation and progression. H19 is one of the most studied lncRNAs involved in BC evolution, which is aberrantly upregulated in human breast tumor tissues and cells and associated with an increased risk of BC [66]. LncRNA HOTAIR is aberrantly expressed in BC compared to the normal mammary epithelium, and its high expression level is correlated with a poor prognosis for BC and an independent biomarker for predicting BC mortality and metastasis [67]. 

Our study emphasizes the importance of unbiased screening approaches and identifies novel signaling pathways that can be regulated via PTHrP. The majority of the differentially expressed lncRNAs in PTHrP WT tumors compared to the KO tumors have not been described previously. The ablation of Pthrp significantly downregulated lncRNA expression, which has the potential to cis-regulate nearby PCGs in both mouse primary BC cells (GM50337 and Scd1) and human BC cell lines (OLMALINC and SCD). Our results indicate that Pthrp-regulated lncRNAs are important mediators during BC progression. Our study provides novel evidence that PTHrP is an important regulator of lncRNA involving fatty acid metabolism and thereby plays a critical role in BC progression. RNAseq analysis often lacks systematic functional validation. Here, we demonstrated the identified PTHrP-associated GM50337 using both computational approaches and molecular assays. We performed an ASO-mediated knockdown of GM50337 and observed a significant reduction in the proliferation of primary mouse BC cells. This suggests that functional lncRNA transcripts contribute to the hallmarks of cancer and, therefore, are appealing potential therapeutic targets. 

However, the functions of lncRNAs in cancer biology are still elusive. Our identification of PTHrP-regulated mouse lncRNA and the study of its relevant functions also have some limitations. First, many lncRNAs in mice are not highly conserved in humans, which often hinders the identification of human orthologs via a mouse model. In our study, we only observed the conservation of mouse GM50337 at the level of genetic location in the human genome but not based on sequencing conservation. Thus, we looked for a potential human ortholog of mouse GM50337 based on its location rather than its sequence conservation. Synteny between lncRNAs and PCGs is frequently conserved across species [68]. Future studies are necessary to unambiguously validate the correct human counterpart. Secondly, we knocked down GM50337 in the mouse primary BC cells and observed a significant decrease in Scd1 expression, indicating that GM50337 has the potential to cis-regulate Scd1. However, deeper functional and structural studies are needed to further elucidate the mechanisms of this GM50337-Scd1 interplay. Such studies could determine whether cis-acting lncRNAs activate the transcription of closely located genes by promoting chromatin looping from transcriptional enhancers [23]. Thirdly, while the lncRNAs mentioned above affect the expression of genes nearby, other lncRNAs act in trans, regulating gene transcription at independent chromosomal loci. Our current study only focused on the lncRNAs with a cis-regulatory function, and future studies will be necessary to explore the lncRNAs with a potential trans-acting function. Finally, the expression of some antisense lncRNAs is downregulated during BC initiation and progression, thus exerting tumor suppressive roles. For example, several human p53-regulated lncRNAs are downregulated in colorectal cancer and in acute lymphocytic leukemia, suggesting their role as tumor suppressors [23]. Here, we only focused on the overexpressed lncRNAs regulated by PTHrP, and future studies will be necessary to explore whether PTHrP has the potential to downregulate lncRNAs, which may have a tumor suppressor function. 

## 5. Conclusions

Our loss-of-function experiments in advanced genetically engineered mouse models of BC progression and in vitro human BC cells demonstrate that PTHrP deletion/inhibition results in a significant BC initiation delay via the interruption of cell growth and fatty acid metabolism remodeling. We propose that targeting these novel pathways, including these lncRNAs controlled by PTHrP, could be used to develop targeted therapeutic strategies in BC and its metastatic complications.

## Figures and Tables

**Figure 1 cancers-15-03763-f001:**
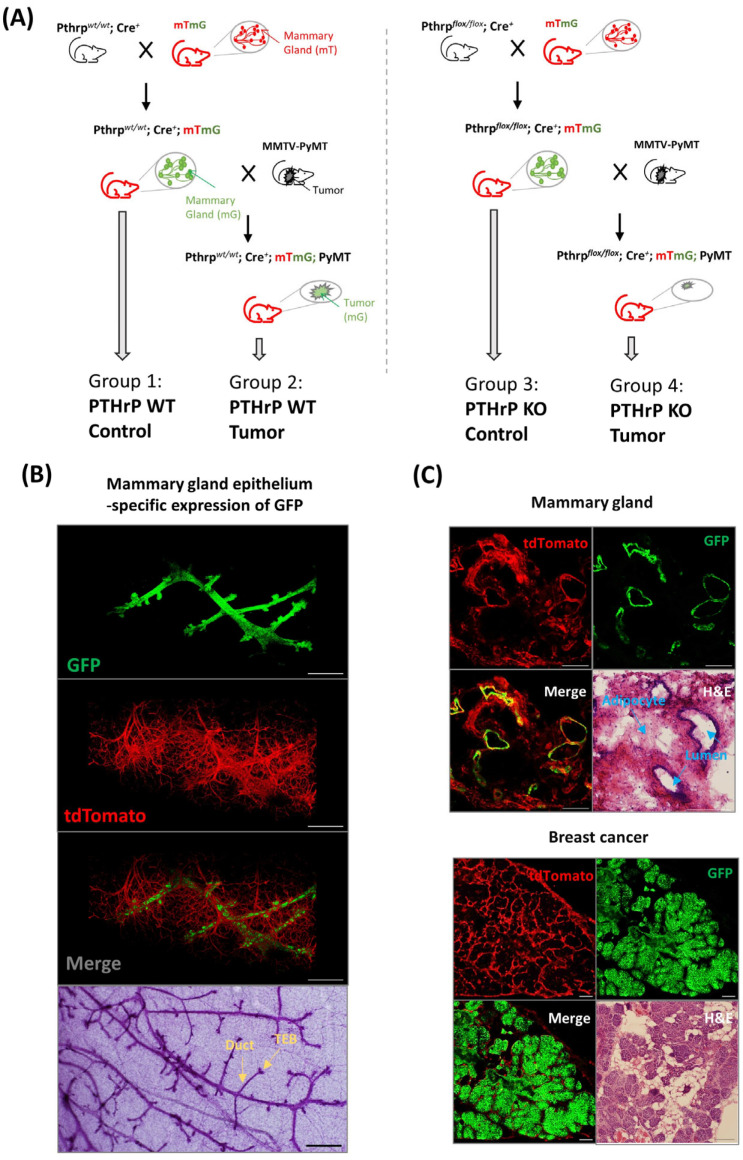
Generation of mouse tumor model. (**A**) Schematic diagram of the genetic construction of four groups of mice. Pthrp*^wt/wt^*; Cre^+^ mTmG tumor-free (Group 1), Pthrp*^wt/wt^*; Cre^+^ mTmG tumor (Group 2), Pthrp*^flox/flox^*; Cre^+^ mTmG tumor-free (Group 3) and Pthrp*^flox/flox^*; Cre^+^ mTmG tumor mice (Group 4). Pthrp*^wt/wt^*; Cre^+^ and Pthrp*^flox/flox^*; Cre^+^ mice were crossed with membrane-targeted Tomato/membrane-targeted GFP (mTmGRosa26 mice) to produce Pthrp*^wt/wt^*; Cre^+^ mTmG tumor-free (Group 1) and Pthrp*^flox/flox^*; Cre^+^ mTmG tumor-free (Group 3), respectively. Cre^+^ mTmG tumor-free (Group 1) and Pthrp*^flox/flox^*; Cre^+^ mTmG tumor-free (Group 3) were further crossed with MMTV-Cre transgenic breast tumor mice to obtain Cre^+^ mTmG tumor (Group 2) and Cre^+^ mTmG tumor mice (Group 4), respectively. (**B**) Ubiquitous expression of mTd and tissue-specific expression of GFP in ducts and alveoli in the mammary gland (fresh tissue). Wholemount staining of mouse mammary glands at the same stages of development showing ductal structures and terminal end buds (TEB) embedded in a fat pad. (**C**) Normal histological appearance of the mammary gland from mTmG; Cre^+^ tumor-free and tumor mice. Upper panel: Mammary gland. H&E staining of cross-section frozen mice mammary gland tissue showing a few normal lumens with epithelial cells. Lower panel: breast cancer. H&E staining of cross-section frozen mice breast cancer tissue.

**Figure 2 cancers-15-03763-f002:**
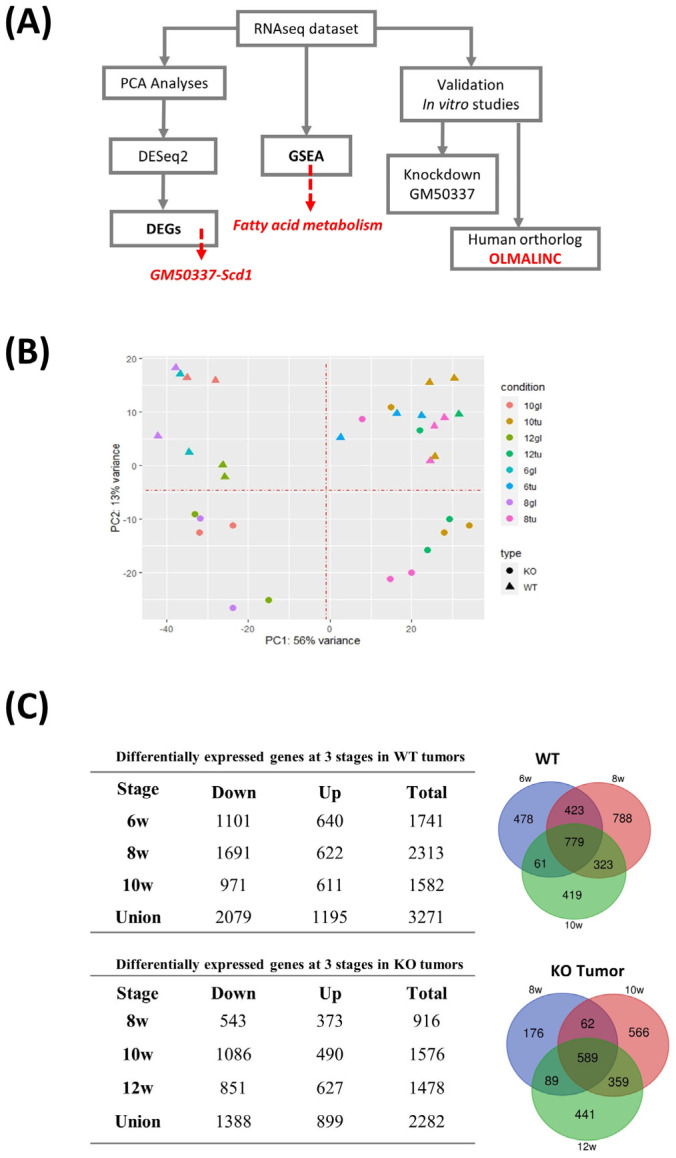
Differentially expressed genes along tumor progression. (**A**) Workflow for RNAseq analyses. (**B**) PCA of the mRNA expression data in tumor and normal samples. (**C**) Differentially expressed genes at 3 stages in WT and KO tumors and Venn diagram of DEGs for PTHrP WT and KO tumors.

**Figure 3 cancers-15-03763-f003:**
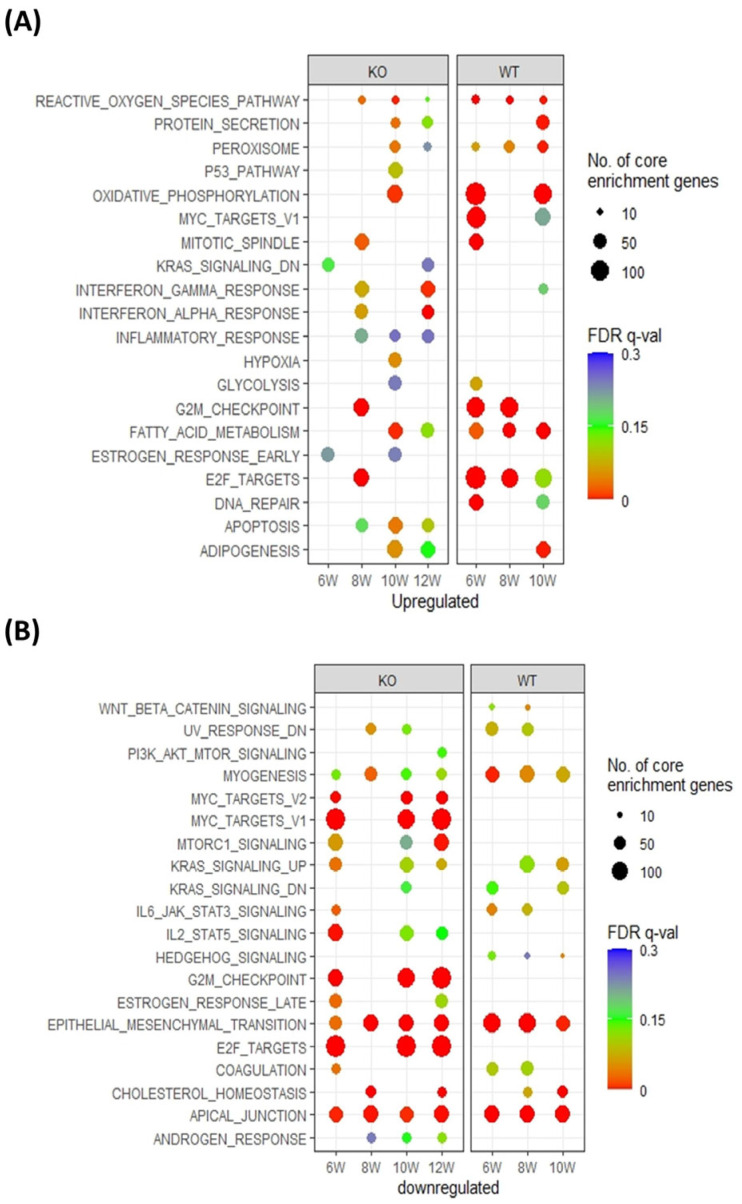
Pathway enrichment analyses. (**A**) Upregulated GSEA-hallmarks in tumor cells compared to tumor-free cells. (**B**) Downregulated GSEA-hallmarks in tumor cells compared to tumor-free cells. GSEA-hallmark enrichment at different stages for PTHrP WT and KO tumors. The top 20 terms with the most significant FDR at each time point were plotted. Color represents –log10 of FDR or overrepresented *p*-value, and dot size represents the number of enrichment genes in each hallmark term.

**Figure 4 cancers-15-03763-f004:**
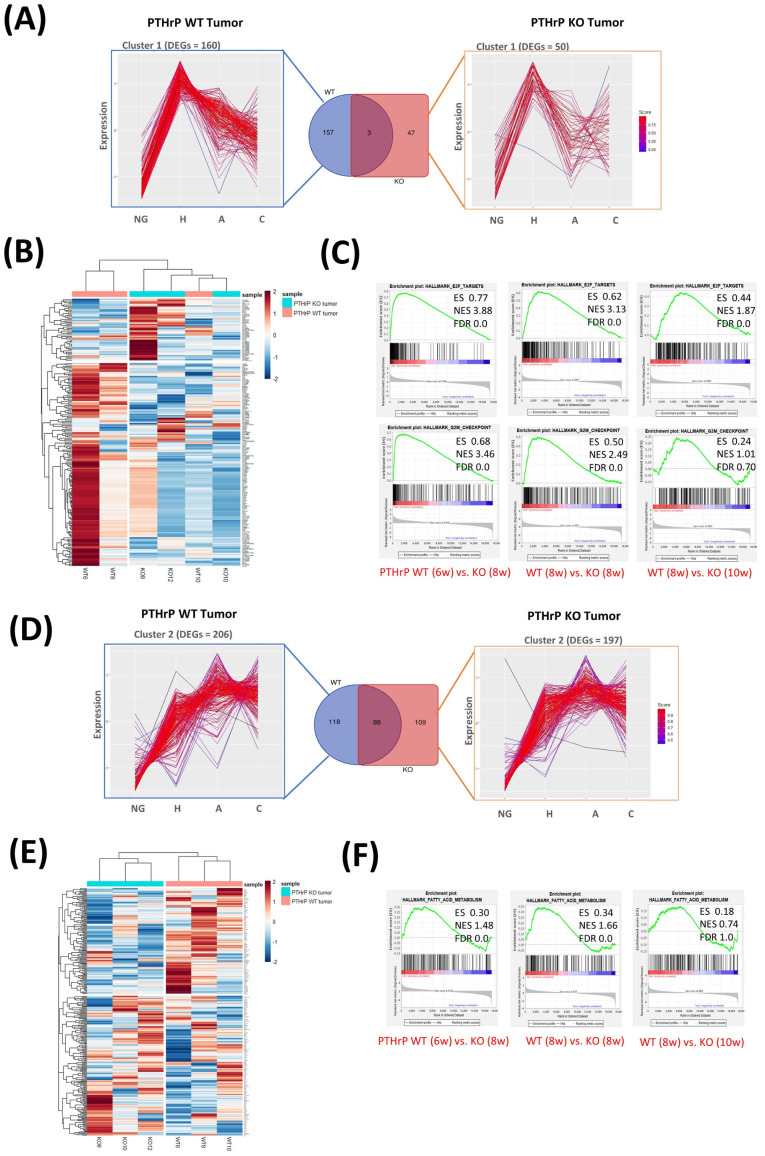
Temporal expression analysis of deregulated genes from WT and KO tumors. (**A**) Cluster 1 of unions of the DEGs from WT and KO tumor at hyperplasia stage. Each line indicates one differential expression gene. The red line indicates the high similar one, and the blue line indicates the low similar one (K = 3). (**B**) RNA-Seq heatmap showing the 202 genes in cluster 1 differentially over-expressed in PTHrP WT at week 6 and KO tumors at week 8 compared to their tumor-free controls, followed by their expression during the tumor development at week 8 and 10 for PTHrP WT and week 10 and 12 for KO tumors, respectively. (*p* value adjusted (padj) < 0.1, log2 fold change (log2FC) > ±1.5). (**C**) GSEA results indicated that PTHrP expression was significantly correlated with the cell cycle-associated gene signatures, including HALLMARK_E2F_TARGETS and HALLMARK_G2M_CHECKPOINT. (**D**) Cluster 2 of unions of the DEGs from WT and KO tumor at hyperplasia stage. Each line indicates one differential expression gene. The red line indicates the high similar one, and the blue line indicates the low similar one (K = 3). (**E**) RNA-Seq heatmap showing the 310 genes in cluster 2 differentially over-expressed in PTHrP WT at week 6 and KO tumors at week 8 compared to their normal controls, followed by their expression during the tumor development at week 8 and 10 for PTHrP WT and week 10 and 12 for KO tumors, respectively. (*p* value adjusted (padj) < 0.1, log2 fold change (log2FC) > ±1.5). (**F**) GSEA results indicated that PTHrP expression was significantly correlated with the fatty acid metabolism-associated gene signatures, including HALLMARK_FATTY_ACID_METABOLISM.

**Figure 5 cancers-15-03763-f005:**
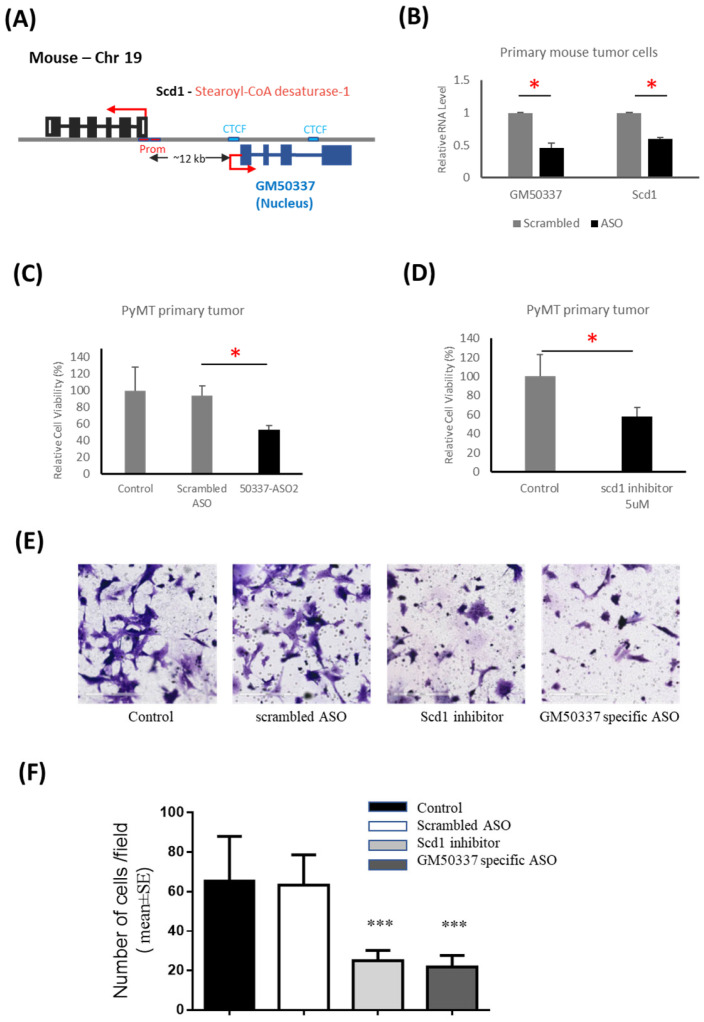
Interruption of GM50337-Scd1 decreases cell viability and proliferation. (**A**) Representation of the GM50337 gene locus. GM50337 is an intergenic lncRNA gene located on mouse chromosome 19, and the GM50337 RNA transcript contains 4 exons and a poly (**A**) tail. The red box indicates promoter region, and the blue box indicates the CTCF binding site. (**B**) qPCR showing GM50337 and Scd1 expression in primary tumor cells after 24 h of incubation with GM50337-specific ASO compared to scASO-treated control cells. Data are presented as mean values ± SD (*n* = 3 independent experiments). * *p* < 0.05 (paired student’s *t*-test; two tailed). (**C**) Primary tumor cells treated with 5 µM ASO target GM50337 for 72 h. A scASO was used as a negative control. Cell viability is normalized to cells treated with scASO. (**D**) Primary tumor cells treated with 5 uM Scd1 inhibitor for 72 h. Cell viability is normalized to untreated cells (‘Control’). Data are presented as mean values ± SD (*n* = 3 independent experiments). * *p* < 0.05 (paired student’s *t*-test; two tailed). (**E**) Matrigel invasion assay. Microscopic picture of invaded cells. Primary tumor cells treated with 5 µM ASO target GM50337 or with 5 µM Scd1 inhibitor for 36 h, respectively. A scASO was used as a negative control. (**F**) Bar graph represents the number of cells that invaded the Matrigel. Values are as mean ± SEM, *n* = 15, *** *p* < 0.001.

**Figure 6 cancers-15-03763-f006:**
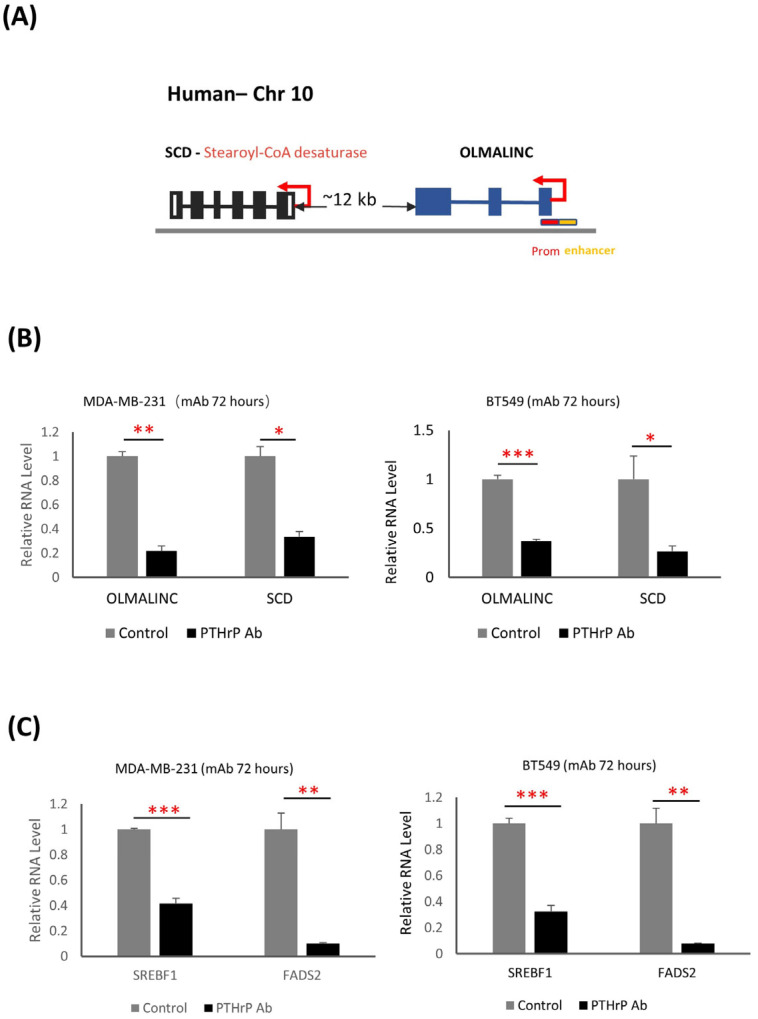
PTHrP mAb treatment to human BC causes a decrease in expression of OLMALINC and target genes. (**A**) Representation of the OLMALINC gene locus. OLMALINC is an intergenic lncRNA gene located on human chromosome 10, and the OLMALINC RNA transcript contains 3 exons and a poly (**A**) tail. The red box indicates promoter region, and the orange box indicates the enhancer region. (**B**) (Left) qPCR showing OLMALINC and SCD expression in MDA-MB-231 after 72 h of incubation with PTHrP Ab compared to the control cells with Ab treatment. (Right) qPCR showing OLMALINC and SCD expression in BT-549 after 72 h of incubation with PTHrP Ab compared to the control cells with Ab treatment. Data are presented as mean values ± SD (*n* = 3 independent experiments). * *p* < 0.05, ** *p* < 0.01, *** *p* < 0.005 (paired student’s *t*-test; two tailed). (**C**) (Left) qPCR showing SREBF1 and FADS2 expression in MDA-MB-231 after 72 h of incubation with PTHrP Ab compared to the control cells with Ab treatment. (Right) qPCR showing SREBF1 and FADS2 expression in BT-549 after 72 h of incubation with PTHrP Ab compared to the control cells with Ab treatment. Data are presented as mean values ± SD (*n* = 3 independent experiments). * *p* < 0.05, ** *p* < 0.01, *** *p* < 0.005 (paired student’s *t*-test; two tailed).

**Table 1 cancers-15-03763-t001:** lncRNA-PCG pair overexpressed in PTHrP WT tumors.

LncRNAs	Protein Coding Genes	PCC	*p* Value	Description	Functions
Up only in PTHrP WT tumors					
A730020E08Rik	Ccser1	0.74	7.05 × 10^−7^	Coiled-Coil Serine Rich Protein 1	Modulate cell division
BC016548	Elf5	0.97	3.14 × 10^−20^	E74 Like ETS Transcription Factor 5	Transcription factor
Gm14133	Syndig1	0.79	4.27 × 10^−8^	Synapse Differentiation Inducing 1	Regulator of excitatory synapse development
Gm26902	Ch25h	0.75	5.07 × 10^−7^	Cholesterol 25-hydroxylase	Enzyme that converts cholesterol to 25-hydroxycholesterol
Gm50337	Scd1	0.91	1.34 × 10^−13^	Stearoyl-CoA desaturase 1	Synthesis of monounsaturated fatty acids
Rps18-ps4	Rps18-ps6	0.71	3.46 × 10^−6^	Ribosomal Protein S18, Pseudogene 6	NA

## Data Availability

The datasets generated and analyzed for RNA-Seq in this publication, corresponding raw read counts matrix, and normalized gene expression matrix reported in this study have been deposited in NCBI’s Gene Expression Omnibus (GEO) [69] and are accessible through GEO Series accession number GSE225877.

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
