# Peer review of "PTHrP Regulates Fatty Acid Metabolism via Novel lncRNA in Breast Cancer Initiation and Progression Models"

_cancers, 2023, doi:10.3390/cancers15153763_

Round 1

Reviewer 1 Report

This is a good article. The purpose is clear, the methods are careful, and all the experiments necessary for the authors' purpose have been made, in the judgment of this reviewer,

The results are good, and the discussion and conclusions are sound, but the authors should make several changes in the presentation.

The panels of figure 1 should be more enlarged, with higher resolution, and the content of the four panels of the two groups in figure 1 should be well explained with more details in the legend. In particular, figure 1C especially needs to be better explained, i.e, allusions to normal histological appearance of the mammary gland from mTmG; Cre+ tumor-free and tumor mice (cross-section frozen tissue)

Both the graphs and the legends in Figure 3 should be explained in more detail, avoiding excessive symbols (ASO, ...) that force the reader to review the entire article to understand their meaning.

Author Response

The panels of figure 1 should be more enlarged, with higher resolution, and the content of the four panels of the two groups in figure 1 should be well explained with more details in the legend. In particular, figure 1C especially needs to be better explained, i.e., allusions to normal histological appearance of the mammary gland from mTmG; Cre+ tumor-free and tumor mice).

We have amended Figure 1 according to the comments: The panels of Figure 1 have been enlarged with higher resolution. We have provided a more detailed new legend especially for panel 1C. Modified Figure 1 and legend (page 309-325) are highlighted.

Reviewer 2 Report

In this manuscript Zhang et al used PTHrP WT/ KO mice to examine the signaling pathways of tumorigenesis from early to late stage. most of the claims are made from the RNA Seq data, where authors found the differential regulation of a novel lncRNA and SCD1. Further, they also validated its role in tumor initiation and progression. Though the manuscript looks interesting and novel. some of the points should be addressed before considering it for publication. 

1. Heatmap of all differentially regulated genes should be added to the main figure with respect to fold change or Z-score. 

2. Pathway analysis could be done with the differentially regulated gene list. (GO, KEGG or BP)

3. A. If authors claim that upregulation of novel lncRNA (GM50337) and SCD1 leads to tumor progression, can depletion of these factor reduce tumor size?

b. Can authors check more parameters to confirm that indeed GM50337 and SCD1 play role in tumorigenesis like colony formation assay, western blots of proliferative markers and in vitro metastasis assay? this would strengthen the manuscript further. 

4. authors should discuss more about lncRNA and cancer in discussion. 

Author Response

Comment #1

Heatmap of all differentially regulated genes should be added to the main figure with respect to fold change or Z-score.).

Our response is as follows: The heatmap requested by the reviewer has been inserted in Figure 4, panels B and E, and the figure legend has been changed accordingly. In addition, the text referring to the heatmap is now included in the result section on page 403-445(highlighted).

Comment #2

Pathway analysis could be done with the differentially regulated gene list.

We have now included a new Figure 3 representing GSEA, showing up- and down-regulated signaling pathways in PTHrP WT compared to KO tumors. Accordingly, in the result section on page 389-396, we added a separate paragraph explaining Figure 3.

Comment #3A

If authors claim that upregulation of novel lncRNA (GM50337) and SCD1 leads to tumor progression, can depletion of these factor reduce tumor size?

We do not know yet if depletion of GM50337 will translate in reduced tumor size. Studies in vivo are planned to examine this possibility. Studies with SCD1 on tumor progression in vivo have been reported previously in several cancer models and have been incorporated in the discussion on page 697-700(highlighted).

Comment #3B

Can authors check more parameters to confirm that indeed GM50337 and SCD1 play role in tumorigenesis like colony formation assay, western blots of proliferative markers and in vitro metastasis assay? this would strengthen the manuscript further.).

We performed in vitro invasion assays to further examine the role of GM50337 and Scd1 in tumorigenesis. These new data have now been inserted in the result section (Figure 5E and 5F with updated legend) on page 539-545(highlighted).Furthermore, we have updated the materials and methods section on page 294-298, accordingly.

Comment #4

Authors should discuss more about lncRNA and cancer in discussion.

Our discussion now includes more details about the role of lncRNAs in cancer progression. Specifically, we discussed the role of lncRNAs in the PyMT breast cancer mouse models on page 719-735.

Round 2

Reviewer 2 Report

The authors have satisfactorily addressed my concerns. I have no other comments.